# PP6 phosphatase and Elongator contribute to kinesin 5-dependent spindle assembly by controlling microtubule regulators levels

Laura Marín[1], Jorge Castro-Sangrador[1⊛], Marta Hoya[1⊛], Shara Tello[1], Pedro M. Coll[1], Javier Encinar del Dedo[1], Alfonso Fernández-Álvarez[1], Juan C. Ribas[1], Phong T. Tran[2,3], Sergio A. Rincon[1]*

1 Instituto de Biología Funcional y Genómica and Departamento de Microbiología y Genética, Consejo Superior de Investigaciones Científicas (CSIC) and Universidad de Salamanca, Salamanca, Spain, 2 Institut Curie, P.S.L. Université, Sorbonne Université, CNRS UMR144, Paris, France, 3 University of Pennsylvania, Cell & Developmental Biology, Philadelphia, Pennsylvania, United States of America

⊛ These authors contributed equally to this work.
* sarpadilla@usal.es

## Abstract

Eukaryotic chromosome segregation relies on the assembly of a bipolar machinery based on microtubules (MTs), named the mitotic spindle. Formation of the mitotic spindle follows a force balance mechanism that ensures the proper capture and separation of sister chromatids. Many proteins have been involved in the establishment of this force balance, although kinesin 5 is well recognized as the major outward pushing force generator, since its inactivation results in monopolar, non-functional spindles. In order to find additional players in the force balance mechanism, we have performed a suppressor screen using a conditional allele of the fission yeast kinesin 5 ortholog Cut7. This screen identified that the lack of the PP6 phosphatase partially suppresses *cut7* phenotypes, at least by defective translation of MT regulators, such as the minus end-directed kinesin Klp2, the MT stabilizer Alp7 and the MT bundler Ase1, impacting on the force balance mechanism. Additionally, our data show that the Elongator complex, a target activated by PP6 for efficient tRNA modification, also contributes to the force balance, albeit to a lesser extent. Importantly, this complex has recently been implicated in direct MT polymerization in metazoans, a role not shared by its fission yeast counterpart.

## Author summary

The mitotic spindle is a cellular machine made of microtubules, which become arranged in a bipolar manner to capture and segregate chromosomes into the daughter cells during cell division. Spindle bipolarization relies on a force balance mechanism established by the function of many proteins, among which,

**Data availability statement:** All relevant data are within the manuscript and its Supporting Information files.

**Funding:** This work has been supported by the grants PID2021-127459NB-I00, PID2021-124971NB-I00 and CNS2022-135153 (Ministerio de Ciencia, Innovación y Universidades, Spain and FEDER, EU, NextGeneration EU, "A way of making Europe"). J.C-S. and S.T. were supported by a predoctoral contract from the Junta de Castilla y León, funded by the European Social Fund Plus. Funders did not play any role in the study design, data collection and analysis, decision to publish, or preparation of the manuscript.

**Competing interests:** The authors have declared that no competing interests exist.

**Abbreviations:** MT, Microtubules; SNP, Single Nucleotide Polymorphism; SAC, Spindle Assembly Checkpoint; SPB, Spindle Pole Body; MBC, Methyl 2-benzimidazolecarbamate; ORF, Open Reading Frame; PAGE, PolyAcrilamide Gel Electrophoresis; PAA, PolyAcrilamide; NGS, Next Generation Sequencing; A600, Absorbance at 600 nm; a.u., Arbitrary Units; n.s., Non Significant

the essential kinesin 5 is the major outward force generator. To discover novel proteins involved in the force balance, we have screened for suppressors of the kinesin 5 ortholog of fission yeast Cut7. Among the hits of the screen, we found that the lack of PP6 phosphatase components allow *cut7* mutants form a bipolar spindle. Our results show that this suppression is, as least in part, mediated by the inactivation of Elongator, a complex that modifies tRNAs to facilitate the translation of specific mRNAs. Our results show that PP6 and Elongator participate in the efficient production of microtubule regulators that contribute to the proper generation of the force balance for spindle assembly.

## Introduction

Mitosis requires the assembly of a MT-based dynamic structure named the mitotic spindle. This cellular machinery has the essential role of capturing the chromosomes and separating the sister chromatids into the daughter cells (reviewed in [1]). Most spindle MTs have their minus ends embedded into the spindle poles. Astral MTs plus ends contact the cell ends to position the mitotic spindle within the dividing cells, kinetochore MTs plus ends probe their environment to catch the chromosome through the kinetochore, a proteinaceous platform that serves as docks for microtubules; finally, interpolar MTs plus ends interdigitate with those of the MTs nucleated from the opposite spindle pole, providing integrity to the bipolar spindle (reviewed in [2]).

Initial stages of spindle assembly are critical for achievement of centrosome separation, also known as bipolarization. Spindle MTs are soon nucleated from the centrosomes after mitotic commitment. These MTs need to be organized in an antiparallel fashion to support spindle bipolarization. The conserved kinesin 5 motors have emerged as the major contributors for spindle bipolarization [3,4]. This kinesin possesses an N-terminal motor head, responsible for the generation of motion upon ATP hydrolysis, a stalk domain, which consists of coiled-coil regions to allow di- and tetra-merization, and a C-terminal regulatory tail [5]. These proteins function as homo-tetramers, in which each dimeric motor is organized antiparallelly, with the motor heads sticking out of the complex. This structure provides kinesin 5 motors the ability to interact with two independent MTs, organizing them in an antiparallel manner, sliding them apart and generating outward pushing forces in the context of the mitotic spindle.

Upon mitotic entry, kinesin 5 accumulates at the spindle poles, enriched in MT minus ends, although it is also detectable as dynamic foci over the spindle length [6,7]. Kinesin 5 MT sliding ability relies on the plus-end directed motion that these motors produce, albeit in fungi they also show minus-end directed motion, responsible for their accumulation at the spindle poles [8–13]. It has been proposed that directionality of this kinesin may be regulated by molecular crowding: single kinesin 5 molecules may move towards the spindle pole, and after clustering, they switch into plus end motility, generating outward pushing forces [12,14]. The localization of

this protein is partly controlled by phospho-regulation of the regulatory C-terminal tail. This region of kinesin 5 contains the conserved BimC box, whose phosphorylation depends on CDK activity, and participates in kinesin 5 targeting onto the spindle MTs [15,16]. This region also contains a NIMA-family kinase box, target of the NIMA kinase Nek6 in human cells, which plays a secondary role in kinesin 5 localization to the spindle [17]. Other studies have uncovered additional CDK-consensus phosphorylation sites in the motor head of the *Saccharomyces cerevisiae* kinesin 5 orthologs, Cin8 and Kip1, that may contribute to overall motor activity and directionality [8,18,19]. Recent works in fission yeast kinesin-5 ortholog, show that multi-phosphorylation of the C-terminal tail may be important for fine-tuning the sliding force produced by this protein for spindle assembly [20].

The lack of a functional kinesin 5 usually results in spindle collapse and unsuccessful chromosome segregation [21–24]. Given the crucial role of kinesin 5 in spindle assembly, this protein has been object of study to develop drugs that specifically inhibit their function in order to avoid uncontrolled cell proliferation [25–27]. Indeed, drugs as monastrol and derivatives as STILC block cell division by acting on kinesin 5, resulting in the formation of monopolar spindles and eventual cell death. However, although chemical inhibition of kinesin 5 results in the formation of monopolar spindles and block of cell division at the population levels, some cells eventually bypass the need for kinesin 5 [28,29].

Proper spindle assembly not only requires outward pushing forces. Indeed, this process requires the establishment of a force balance that is modulated as mitosis proceeds. Minus-end directed motors, dynein and kinesin 14 have emerged as the major producers of kinesin 5 counteracting forces. In fact, cells lacking kinesin 5 are not capable of spindle bipolar formation, in part, due to excessive inward pulling forces produced by dynein and/or kinesin 14 [30–32].

In *Schizosaccharomyces pombe*, the only kinesin 5 ortholog is Cut7, which is essential for spindle assembly. It has been established that the absence of the kinesin 14 Pkl1 allows *cut7Δ* cells to assemble a bipolar spindle [33–36], which, despite showing a shorter length before anaphase onset and delayed anaphase entry, is fully competent for chromosome segregation. This proves that spindle assembly can be achieved in the absence of kinesin 5. In these conditions, a number of MT-associated proteins become essential for bipolar spindle formation, such as the MT polymerizing proteins Alp7/TACC and Alp14/ chTOG, or the MT antiparallel bundler Ase1/PRC1 [35,36].

Previous works have established the importance of kinesin 14 and MT dynamics in the context of Cut7 compromised function [36,37]. Importantly, the ability of the *cut7* thermo-sensitive mutant *cut7–22* (which harbors the mutation P1021S at the C-terminal tail, close to phosphorylation sites [20], to assemble a bipolar spindle showed a good anticorrelation with the levels of the other kinesin 14 ortholog, Klp2 at the mitotic spindle. Additionally, decreased MT stability also favored bipolar spindle formation in *cut7–22* cells at restrictive temperature, possibly by inhibiting kinesin 14 function and favoring the force balance into a net pushing force.

In this work, we have performed a suppressor screen trying to find novel players involved in the force balance mechanism for spindle assembly. Interestingly, we found that the absence of PP6 phosphatase components, both the catalytic Ppe1 and the regulatory Ekc1 subunits, allow spindle assembly upon Cut7 inactivation. This phosphatase complex has been identified as a positive regulator of the Elongator complex [43]. This multimeric complex is responsible for the tRNA modification when a uridine is present in the wobble position, stabilizing the codon – anticodon interaction and increasing the robustness of protein translation whose mRNAs are enriched in AAA (coding for lysine) or GAA (coding for glutamate) codons. Interestingly, we also found that the absence of a functional Elongator complex also allows bipolar spindle formation in *cut7* thermo-sensitive mutants. The suppression of the *cut7* mutant thermo-sensitivity is fully dependent on Elongator catalytic activity, and not on direct MT regulation, ruling out a role for this complex in MT dynamics control in fission yeast, in contrast with the role of Elongator in MT polymerization in metazoans. Finally, we propose that Ppe1 has additional roles in protein translation regulation independent of the regulation of the Elongator complex, which may account for the more robust suppression of the *cut7* mutant phenotype observed in cells lacking PP6 than the one shown by cells lacking Elongator components.

## Results

### The absence of PP6 phosphatase allows bipolar spindle assembly in kinesin 5/Cut7 mutants

The only ortholog of the kinesin 5 protein is Cut7 in fission yeast. Cut7 is an essential protein, required for bipolar spindle assembly by the generation of outward pushing forces [21]. We and others have demonstrated that fission yeast cells can assemble a functional mitotic spindle when both Cut7 and the opposing kinesin 14, Pkl1, are absent [33–36]. In order to find novel proteins involved in the control of this force balance mechanism, we performed a suppressor screen using the *cut7* conditional allele *cut7–24*, which contains the mutation V968A close to the C-terminal BimC box, involved in kinesin regulation [38]. This mutant cannot assemble a bipolar spindle at the restrictive temperature of 36°C. Briefly, *cut7–24* cells were exposed to UV radiation to induce additional mutations (see Material and Methods section for details), and incubated at restrictive temperature to isolate clones capable of forming colonies in these conditions. Genomic DNA from the 28 suppressors obtained in the screen was purified and sequenced to identify the additional mutations responsible for the suppression of the thermo-sensitivity. Interestingly 5 of these clones harbored SNPs (Single Nucleotide Polymorphisms) in *ppe1* ORF and 1 of them contained a mutation in *ekc1* ORF, which code for the catalytic and regulatory subunits, respectively, of the PP6 phosphatase in *S. pombe*. Other interesting suppressors contained SNPs in the genes coding for the Polo-like kinase Plo1 or the inner nuclear membrane protein Man1. For a complete list of the results from the screen, see S2 Table. Given the high frequency of appearance of PP6 phosphatase mutants in our screen, we decided to study its role in the suppression of the *cut7–24* phenotype.

First of all, we confirmed the result of the screen by using the full deletion of *ppe1* or *ekc1*. Indeed, the absence of any of the PP6 phosphatase subunits partially suppressed the thermo-sensitivity of *cut7–24* cells (Figs 1A and S1A). Another thermo-sensitive allele of Cut7, *cut7–22*, was used to confirm this result. Given that the suppression of the phenotype was more robust in the *cut7–22* allele (Fig 1A), we decided to continue this study using primarily this allele. The ability of the double mutant *cut7–22 ppe1Δ* to grow at restrictive temperature, suggested that these cells are capable of bipolar spindle assembly in these conditions. To verify this, we imaged through time-lapse fluorescent microscopy wild type, *ppe1Δ*, *cut7–22* and *cut7–22 ppe1Δ* expressing the Spindle Pole Body (SPB) marker Sid4 fused to the mCherry fluorescent protein at restrictive temperature to analyze mitotic spindle formation (Fig 1B). As expected, contrary to the single mutant *cut7–22*, the majority of *cut7–22 ppe1Δ* cells succeeded in bipolar spindle formation (20% vs. 80%, respectively, Fig 1C). We next analyzed spindle dynamics in wild type, *ppe1Δ* and *cut7–22 ppe1Δ* cells. *cut7–22* cells were not included in this analysis, given that most of the cells produced a monopolar spindle and eventually lysed upon constriction of the cytokinetic ring on the dividing nucleus. *cut7–22 ppe1Δ* cells showed a slight delay in anaphase entry, in an otherwise successful spindle elongation (Fig 1D).

### Cut7 and Pkl1 behave normally in the absence of Ppe1

Given that Cut7 is the major producer of outward pushing forces during spindle formation, we wondered if PP6 phosphatase might be involved in the direct control of this kinesin at mitotic entry, during spindle bipolarization. Cut7 shows a nucleoplasmic localization during interphase followed by a rapid accumulation at the SPBs upon mitotic entry [22]. To check our hypothesis, we analyzed Cut7 recruitment to the SPBs until metaphase in wild type and *ppe1Δ* cells expressing both Cut7 fused to the green fluorescent protein variant Envy [39] and the α tubulin subunit Atb2 fused to mCherry. Time 0 was determined by the appearance of spindle microtubules. Study of Cut7-Envy intensity during spindle bipolarization in wild type and *ppe1Δ* cells did not show significative differences (Fig 1E, 1F), suggesting that Cut7 may not be a target of PP6 phosphatase. Indeed, no differences in Cut7 levels in wild type and *ppe1Δ* cells could be detected by western blot (S1B, S1C Fig). In order to rule out the possibility that PP6 phosphatase may be dephosphorylating Cut7, we studied Cut7 migration by PAGE in the presence of PhosTag, which slows down migration of phosphorylated proteins [40]. Since Cut7 essential function takes place during spindle assembly, we focused on mitotic cells. To do so, we obtained total protein

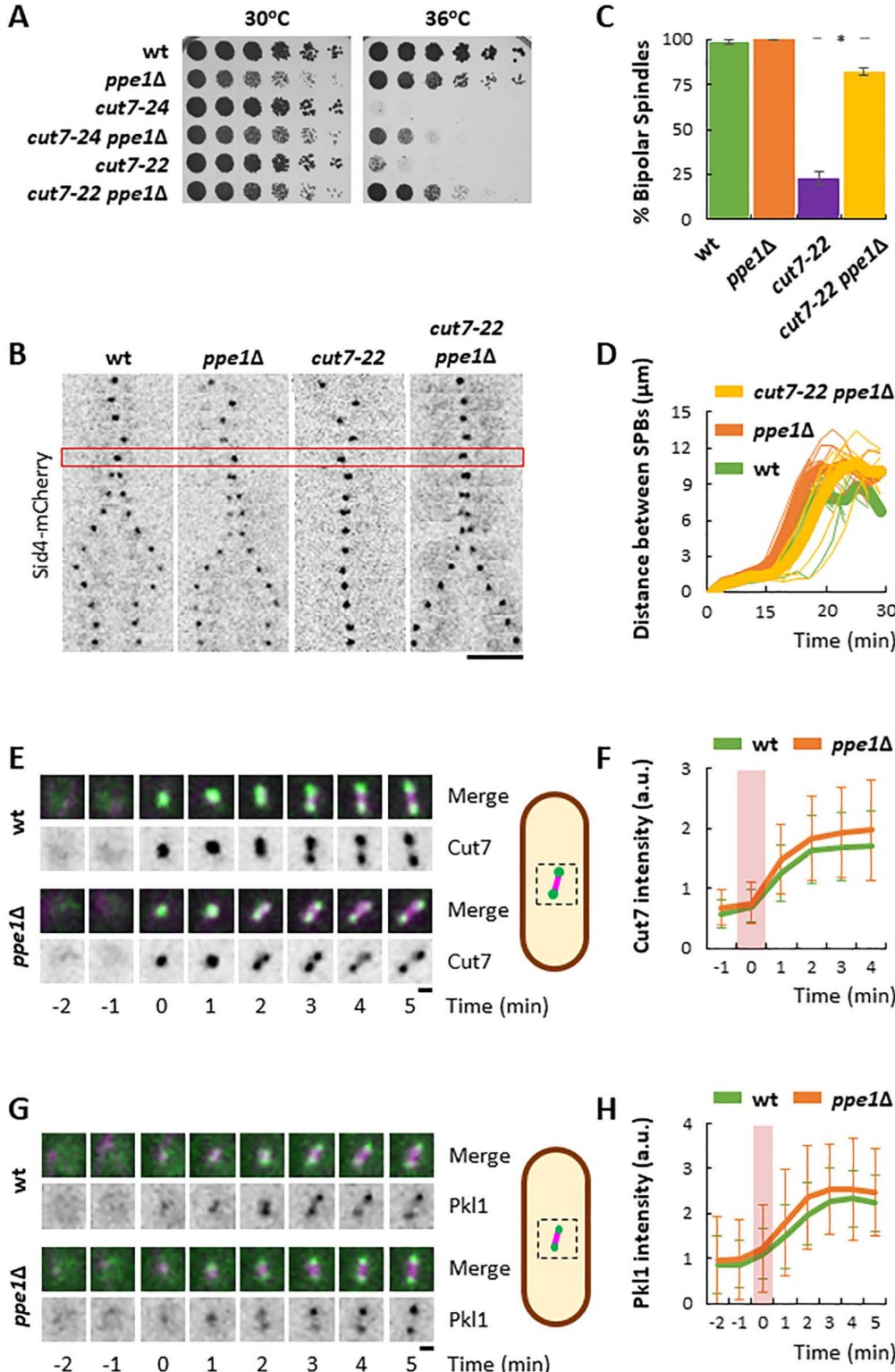

**Fig 1. The absence of PP6 activity suppresses the inability of *cut7* mutants to assemble a bipolar spindle.** A Serial dilution assay of the wild type, *ppe1Δ*, *cut7–24*, *cut7–24 ppe1Δ*, *cut7–22* and *cut7–22 ppe1Δ* strains incubated for 3 days at the indicated temperatures. B: Kymographs of time-lapse fluorescent microscopy showing the spindle dynamics of the wild type, *ppe1Δ*, *cut7–22* and *cut7–22 ppe1Δ* strains expressing Sid4-mCherry. The

red rectangle depicts mitotic entry, which was established by the lack of movement of the SPB. Each frame corresponds to 2 minutes. Scale bar: 5 µm. C: Plot showing the average percentage of cells that managed to assemble a bipolar spindle in three independent experiments. n for wild type: 206; n for *ppe1Δ*: 184; n for *cut7–22*: 236; n for *cut7–22 ppe1Δ*: 283. Statistical difference between *cut7–22* and *cut7–22 ppe1Δ* was determined using a T-test (p = 4 x 10⁻⁴). D: Plot depicting the dynamics of spindle formation. Bold lines represent the average of the spindle dynamics for each strain. n for wild type: 10; n for *ppe1Δ*: 9; n for *cut7–22 ppe1Δ*: 9. **E:** On the left, time lapse images of a magnification of the nuclear region of wild type and *ppe1Δ* cells, expressing Cut7-Envy and mCherry-Atb2 incubated at 25°C. Maximum intensity projections of confocal images are shown. The upper series shows the magenta (mCherry-Atb2) and green (Cut7-Envy) merged channels. The bottom series shows the inverted green channel in a grey scale. Time 0 corresponds to mitotic entry. Scale bar: 1 µm. On the right, a schematic representation of the region of the cell analyzed on the left. F: On the right, plot showing the average accumulation of the Cut7-Envy signal intensity at the spindle region upon mitotic entry. The pink rectangle on the graph depicts the appearance of spindle MTs. n for wild type: 30; n for *ppe1Δ*: 30. a.u.: arbitrary units. Statistical significance was determined using a Repeated Measures ANOVA (p = 0.151). G: On the left, time lapse images of a magnification of the nuclear region of wild type and *ppe1Δ* cells, expressing Pkl1-neonGreen and mCherry-Atb2 incubated at 25°C. Maximum intensity projections of confocal images are shown. The upper series shows the magenta (mCherry-Atb2) and green (Pkl1-neonGreen) merged channels. The bottom series shows the inverted green channel in a grey scale. Time 0 corresponds to mitotic entry. Scale bar: 1 µm. On the right, a schematic representation of the region of the cell analyzed on the left. H: Plot showing the average accumulation of the Pkl1-neonGreen signal intensity at the spindle region upon mitotic entry. The pink rectangle on the graph depicts the appearance of spindle MTs. n for wild type: 22; n for *ppe1Δ*: 25. a.u.: arbitrary units. Statistical significance was determined using a Repeated Measures ANOVA (p = 0.360).

extracts from *nda3-KM311* wild type and *ppe1Δ* cells expressing Cut7-3xHA after 7 hours incubation at 18°C to allow mitotic synchronization. Our results showed no differences in Cut7 migration in wild type and *ppe1Δ* cells, both during the mitotic block or asynchronous growth (S1D Fig), reinforcing the idea that Cut7 may not be under direct control of PP6 phosphatase.

The other major player in the force balance mechanism for spindle formation is the minus end-directed kinesin 14 Pkl1 in *S. pombe*. We first compared Pkl1 localization in wild type and *ppe1Δ* cells. To do this, we imaged both strains expressing Pkl1 fused to the green fluorescent protein neonGreen [41] and mCherry-Atb2. Pkl1 has been shown to localize to the SPB at the early stages of spindle formation, where it contributes to focusing MT minus ends at the spindle poles [34,42]. Our analysis shows a similar pattern of accumulation of Pkl1-nGreen upon mitotic entry in both wild type and *ppe1Δ* cells (Fig 1G, 1H), suggesting that Pkl1 is not regulated by PP6 phosphatase. In fact, the analysis of Pkl1 global levels in both strains by western blot confirmed this result (S1E, S1F Fig). Similar to Cut7, we studied Pkl1 migration in PhosTag SDS-PAGE, by obtaining protein extracts from *nda3-KM311* blocked wild type and *ppe1Δ* cells. No major differences in the migration pattern of Pkl1 were detected in these conditions (S1G Fig), suggesting that PP6 phosphatase does not regulate Pkl1 for bipolar spindle assembly. Collectively, these results suggest that PP6 phosphatase contributes to bipolar spindle assembly independently of both Cut7 and Pkl1.

### The absence of the Elongator complex contributes to bipolar spindle assembly in kinesin 5/Cut7 mutants

Ppe1 has been suggested as the phosphatase that counteracts the inhibitory phosphorylation of the Elongator complex performed by the kinase Gsk3 [43]. Elongator is an evolutionarily conserved complex involved in the methyl-carbonylation of tRNAs in which a uridine is present in the wobble position (U34) [44]. This modification is necessary to overcome the reduced efficiency during translation, a consequence of the low effective interactions between the adenine in the third position of the codon from the mRNA and the U34 from the tRNA. This results in reduced protein translation efficiency in mRNAs enriched in AAA (coding for lysine) and GAA (coding for glutamate) [45]. The Elongator complex is formed by two copies of the subcomplex Elp1-Elp2-Elp3, which form a symmetric scaffold, and bind to one copy of the subcomplex Elp4-Elp5-Elp6 [46,47]. Although every subunit from the complex is required for tRNA modification, Elp3 contains the catalytic activity [48]. *In vitro* experiments have shown that Elp4 is the target of Gsk3 kinase, which phosphorylates S144 of Elp4 to inhibit Elongator catalytic activity [43].

Therefore, we hypothesized that the suppression of *cut7* mutant thermo-sensitivity by the lack of PP6 phosphatase might be the result of impaired activation of the Elongator complex. To test this, we first analyzed if the absence of a functional Elongator complex suppressed the *cut7* mutant phenotype. Indeed, deletion of either *elp3* or *elp4* suppressed

the inability of *cut7–22* cells to assemble a bipolar spindle and restored growth at restrictive temperature to a small extent (Figs 2A-C and S2A). Interestingly, recent studies have shown that, independently of its tRNA-modifying activity, the Elongator complex directly binds to MTs, promoting their growth, inhibiting their catastrophe and ensuring asymmetric central spindle organization for asymmetric cell division in *Drosophila* sensory organ precursor cells [49]. In order to study a possible role of Elongator in direct MT regulation in fission yeast, we analyzed MT dynamics in interphasic wild type, *ppe1Δ*, *elp3Δ* and the catalytically inactive *elp3-YY527AA*. The *elp3-YY527AA* mutant harbors alanine-substitution mutations on both of the conserved tyrosine residues Y527A and Y528A, known to be required for the acetyl and methyl transferase activity of the Elongator complex, but retaining its ability to control MT stability *in vitro* [49,50]. Our results suggest that neither *elp3* nor *ppe1Δ* mutant cells show major defects in MT dynamics (S2B-E Fig). Indeed, contrary to the situation

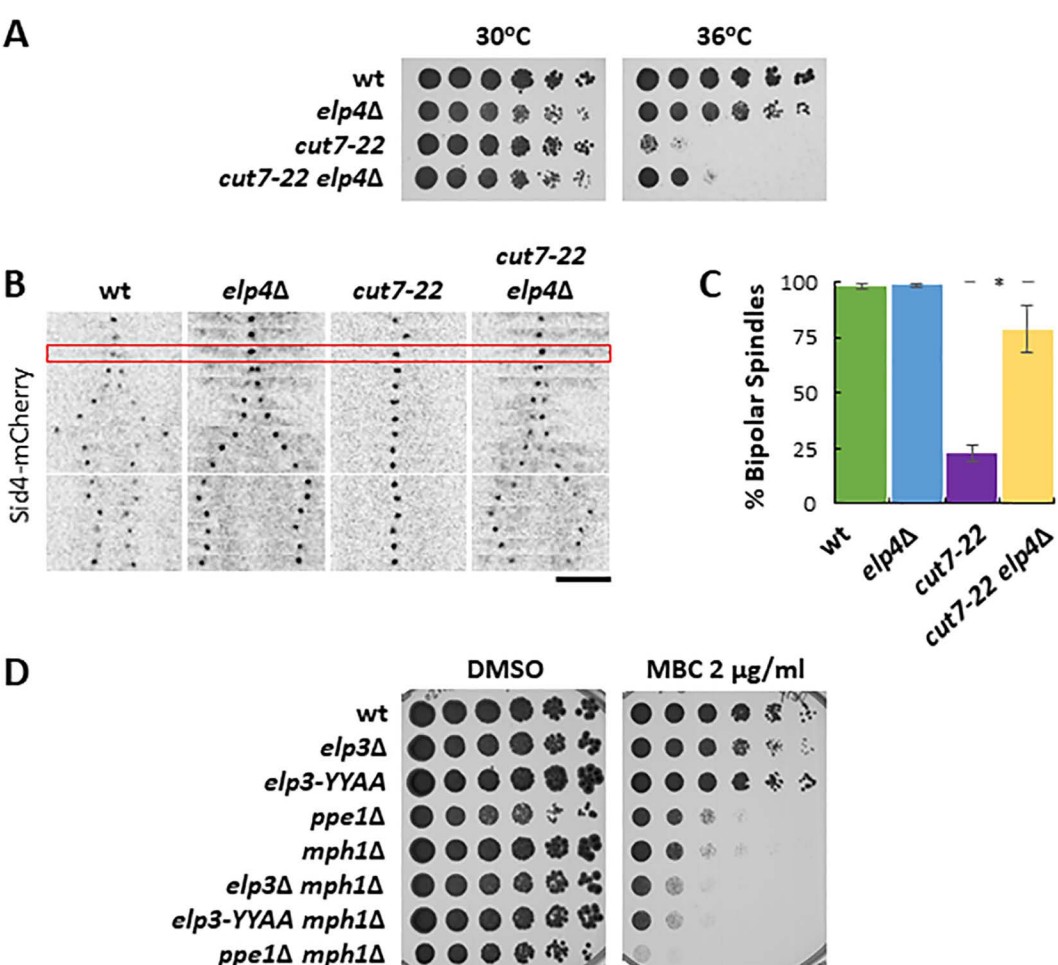

**Fig 2. The absence of Elongator complex components rescues the ability of *cut7* mutants to assemble a bipolar spindle and compromises mitotic progression upon SAC inhibition.** A: Serial dilution assay of the wild type, *elp4Δ*, *cut7–22* and *cut7–22 elp4Δ* strains incubated for 3 days at the indicated temperatures. B: Kymographs showing the spindle dynamics of the wild type, *ppe1Δ*, *cut7–22* and *cut7–22 ppe1Δ* strains expressing Sid4-mCherry. The red rectangle depicts mitotic entry, which was established by the lack of movement of the SPB. Each frame corresponds to 2 minutes of elapsed time. Scale bar: 5 μm. C: Plot showing the percentage of cells that managed to assemble a bipolar spindle in at least three independent experiments. n for wild type: 206; n for *elp4Δ*: 168; n for *cut7–22*: 236; n for *cut7–22 elp4Δ*: 325. Statistical difference between *cut7–22* and *cut7–22 elp4Δ* was determined using a T-test (p = 2 x 10⁻⁴) D: Serial dilution assay of the wild type, *elp3Δ*, *elp3-YY527AA*, *ppe1Δ*, *mph1Δ*, *mph1Δ elp3Δ*, *mph1Δ elp3-YY527AA*, and *mph1Δ ppe1Δ* strains incubated for 5 days at 28°C in the presence of the indicated concentrations of MBC.

in *Drosophila*, Elp3 or Elp4 fused to GFP did not localize to MTs at any stage of the cell cycle in fission yeast cells (S2C Fig). Western-blot analysis confirmed the presence of Elp3-GFP and Elp4-GFP in cell extracts (S2F-G Fig), hence, the lack of localization to MTs is not due to protein degradation. Altogether, these results suggest that the suppression of the *cut7* phenotype by the absence of a functional Elongator complex is the consequence of defective translation of AAA and GAA-enriched mRNAs rather than impaired direct MT regulation.

Given that the absence of both PP6 phosphatase and the Elongator complex suppressed the kinesin 5 mutant *cut7–22*, we wondered if they played specific roles during mitotic progression. Although *ppe1Δ* cells do not show major defects in spindle dynamics (Fig 1D), to unveil genetic interactions, we combined *ppe1Δ* and *elp3* mutants to *mph1*-deleted cells, which lack the major kinase that organizes the surveilling Spindle Assembly Checkpoint (SAC). The SAC is a kinetochore-based signaling pathway that monitors proper MT-kinetochore contacts and in *S. pombe* the SAC becomes essential only when mitosis is compromised [51]. Therefore, a serial dilution assay using single and double mutants in the presence of the MT-depolymerizing drug MBC was performed. Our results show that both *elp3Δ* and *elp3-YY527AA* mutants, and especially *ppe1Δ* cells are very sensitive to MBC when the SAC function is compromised (Fig 2D). This result points towards a role of Ppe1 and Elongator in the control of mitosis progression. Altogether, these results suggest that the Elongator complex does not directly regulate MT dynamics in fission yeast; however, both PP6 phosphatase and Elongator may indirectly control MT cytoskeleton by ensuring effective translation of MT regulators.

### The levels of MT regulators involved in the force balance mechanism for bipolar spindle assembly are reduced in the absence of both functional PP6 phosphatase and Elongator complex

Although Cut7 and Pkl1 are major actors in bipolar spindle assembly, other proteins have been found to participate in the process, ensuring successful mitosis. In particular, previous studies established that Klp2, the other kinesin 14 ortholog in fission yeast, together with proteins involved in MT nucleation/ stability, as Alp7, play an important role in bipolar spindle assembly, which becomes more evident when Cut7 function is compromised [37]. Since the suppression of the *cut7* mutant phenotypes that result from the lack of PP6 phosphatase or a functional Elongator complex does not seem to derive from a major defect in MT dynamics, we wondered if it could stem from faulty protein translation. Therefore, we decided to check the protein levels of those factors whose absence were shown to rescue *cut7* mutant thermo-sensitivity.

We first analyzed the total levels of Klp2, responsible for the inward pulling forces that collapse *cut7* mutant spindles, during the first 3 minutes of mitotic entry, when spindle bipolarization is taking place. To do this, we performed time-lapse imaging on wild type, *ppe1Δ* and *elp3Δ* cells expressing Klp2 fused to neonGreen and mCherry-Atb2. Klp2-nGreen localizes along the interphasic MTs and to weak puncta along the pre-anaphasic spindle [37,52,53]. Since Klp2 levels are too weak to perform image quantification on the spindle over the time, we analyzed the average levels of Klp2-neonGreen within the first 3 minutes after spindle MT nucleation was detected. Our analysis shows a mild, but significant reduction in the total levels of this protein in *elp3Δ* cells, which becomes stronger in *ppe1Δ* cells (Fig 3A, 3B). To confirm this result, we analyzed Klp2 levels by western-blot in wild type, *ppe1Δ* and *elp3Δ* asynchronic cells expressing Klp2 fused to 13xMyc. A similar reduction in Klp2 levels was also detected (Fig 3C, 3D), suggesting that the reduction in Klp2 levels is not exclusive to spindle assembly. Indeed, *klp2* transcript has a codon bias for AAA versus AAG for lysine (66% vs. 34%) and for GAA versus GAG for glutamate (74% vs. 26%), which, altogether, may explain the reduction in the amounts of Klp2, also confirmed by a previous study, which showed a mild decrease of Klp2 levels in Elongator defective cells [54]. The analysis of the genetic interaction between *cut7–22*, *ppe1Δ* and *klp2Δ* showed that the suppression of the thermo-sensitivity of *cut7–22* by the lack of Ppe1 or Klp2 is not additive (S3A Fig). This suggests that the *cut7–22* suppression by *ppe1Δ* may be due, at least in part, to the reduced levels of Klp2 in cells lacking PP6 phosphatase.

A similar approach was performed with the MT-stabilizing protein Alp7, ortholog of the Transforming Acidic Coiled-Coil protein, TACC. Similar to *klp2*, *alp7* mRNA has a skew codon usage for AAA versus AAG for lysine (68% vs. 32%) and for GAA versus GAG for glutamate (60% vs. 40%). Alp7 accumulates at the SPBs upon mitotic entry and

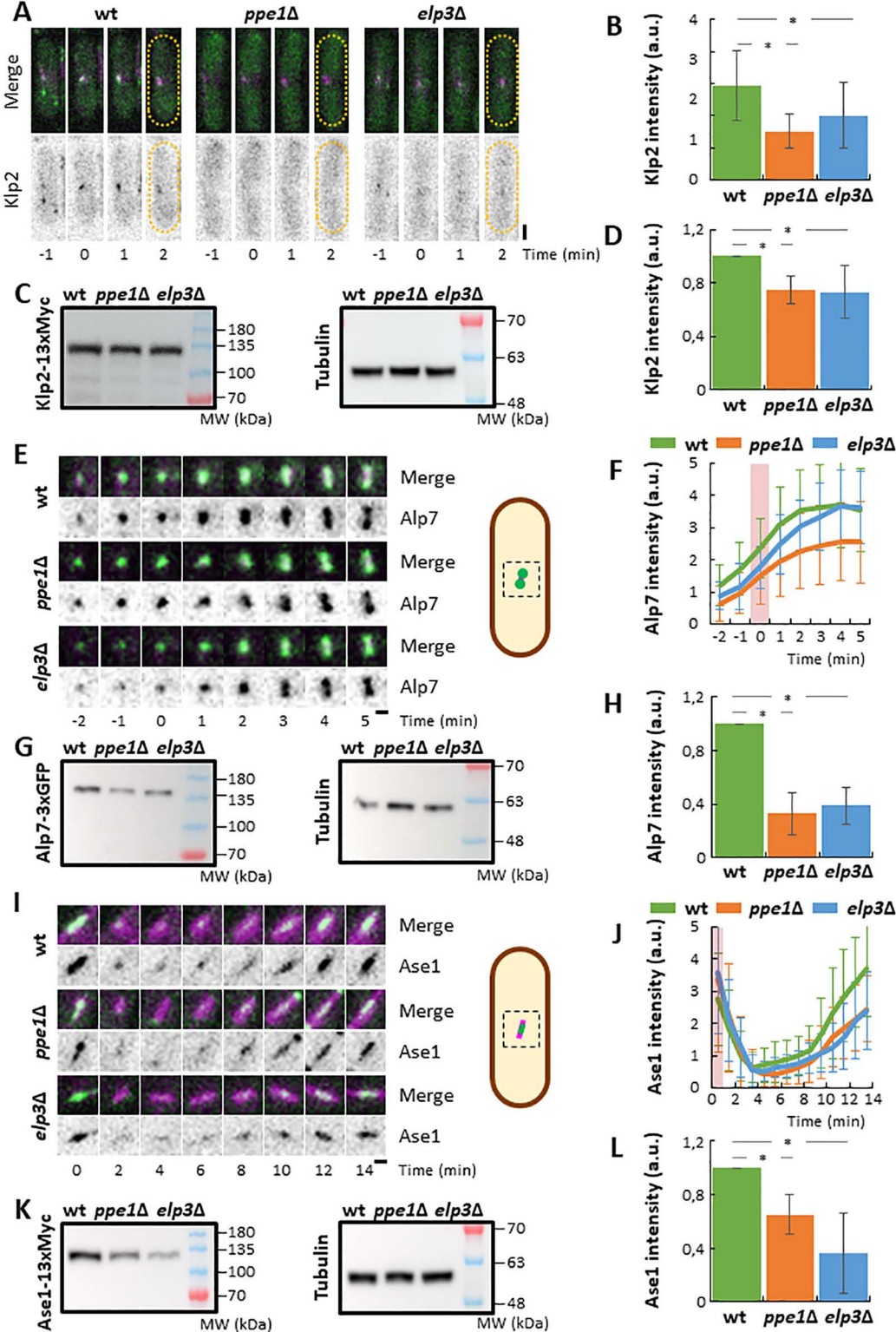

**Fig 3. The accumulation of Klp2, Alp7 and Ase1 to early mitotic spindles is reduced in *ppe1Δ* and *elp3Δ* cells.** A: Time lapse images of wild type, *ppe1Δ* and *elp3Δ* cells, expressing Klp2-neonGreen and mCherry-Atb2 incubated at 25°C. Maximum intensity projections of confocal images are shown. The upper series shows the magenta (mCherry-Atb2) and green (Klp2-neonGreen) merged channels. The bottom series shows the inverted green

channel in a grey scale. Time 0 corresponds to mitotic entry. The dashed line represents the cell contour. Scale bar: 2 µm. B: Plot showing the average of the Klp2-neonGreen signal intensity at the spindle region during the first three minutes upon mitotic entry. n for wild type: 16; n for $ppe1\Delta$: 14; n for $elp3\Delta$: 11. a.u.: arbitrary units. Statistical differences were determined using a T-test (wt vs. $ppe1\Delta$, $p = 1 \times 10^{-4}$; wt vs. $elp3\Delta$, $p = 0.037$). C: Western-blot analysis of SDS PAGE of Klp2-13xMyc levels on wild type, $ppe1\Delta$ and $elp3\Delta$ cells. The left panel corresponds to the blot incubated with an anti-Myc antibody, while the right panel corresponds to the blot incubated with an anti-tubulin antibody. D: Plot showing the total Klp2-13xMyc levels relative to the corresponding tubulin levels in wild type, $ppe1\Delta$ and $elp3\Delta$ cells. n for wild type: 4; n for $ppe1\Delta$: 4; n for $elp3\Delta$: 4. a.u.: arbitrary units. Statistical differences were determined using a T-test (wt vs. $ppe1\Delta$, $p = 0.008$; wt vs. $elp3\Delta$, $p = 0.037$). E: On the left, time lapse images of a magnification of the nuclear region of wild type, $ppe1\Delta$ and $elp3\Delta$ cells, expressing Alp7-neonGreen and mCherry-Atb2 incubated at 25°C. Maximum intensity projections of confocal images are shown. The upper series shows the magenta (mCherry-Atb2) and green (Alp7-neonGreen) merged channels. The bottom series shows the inverted green channel in a grey scale. Time 0 corresponds to mitotic entry. Scale bar: 1 µm. On the right, a schematic representation of the region of the cell shown on the left. F: Plot showing the average accumulation of the Alp7-neonGreen signal intensity at the spindle region upon mitotic entry. The pink rectangle on the graph depicts the appearance of spindle MTs. n for wild type: 51; n for $ppe1\Delta$: 40; n for $elp3\Delta$: 26. a.u.: arbitrary units. Statistical differences were determined using a Repeated Measures ANOVA (wt vs. $ppe1\Delta$, $p = 7 \times 10^{-6}$; wt vs. $elp3\Delta$, $p = 0.127$). G: Western-blot analysis of SDS PAGE of Alp7-3xGFP levels on wild type, $ppe1\Delta$ and $elp3\Delta$ cells. The left panel corresponds to the blot incubated with an anti-GFP antibody, while the right panel corresponds to the blot incubated with an anti-tubulin antibody. H: Plot showing the total Alp7-3xGFP levels relative to the corresponding tubulin levels in wild type, $ppe1\Delta$ and $elp3\Delta$ cells. n for wild type: 3; n for $ppe1\Delta$: 3; n for $elp3\Delta$: 3. a.u.: arbitrary units. Statistical differences were determined using a T-test (wt vs. $ppe1\Delta$, $p = 0.009$; wt vs. $elp3\Delta$, $p = 0.008$). I: On the left, time lapse images of wild type, $ppe1\Delta$ and $elp3\Delta$ cells, expressing Ase1-GFP and mCherry-Atb2 incubated at 25°C. Maximum intensity projections of confocal images are shown. The upper series shows the magenta (mCherry-Atb2) and green (Ase1-GFP) merged channels. The bottom series shows the inverted green channel in a grey scale. Time 0 corresponds to mitotic entry. Scale bar: 1 µm. On the right, a schematic representation of the region of the cell analyzed on the left. J: Plot showing the average accumulation of the Ase1-GFP signal intensity at the spindle region upon mitotic entry. The pink rectangle on the graph depicts the appearance of spindle MTs. n for wild type: 56; n for $ppe1\Delta$: 59; n for $elp3\Delta$: 30. a.u.: arbitrary units. Statistical differences were determined using a Repeated Measures ANOVA (wt vs. $ppe1\Delta$, $p = 3 \times 10^{-4}$; wt vs. $elp3\Delta$, $p = 6 \times 10^{-3}$). K: Western-blot analysis of SDS PAGE of Ase-13xMyc levels on wild type, $ppe1\Delta$ and $elp3\Delta$ cells. The left panel corresponds to the blot incubated with an anti-Myc antibody, while the right panel corresponds to the blot incubated with an anti-tubulin antibody. L: Plot showing the total Ase1-13xMyc levels relative to the corresponding tubulin levels in wild type, $ppe1\Delta$ and $elp3\Delta$ cells. n for wild type: 3; n for $ppe1\Delta$: 3; n for $elp3\Delta$: 3. a.u.: arbitrary units. Statistical differences were determined using a T-test (wt vs. $ppe1\Delta$, $p = 0.028$; wt vs. $elp3\Delta$, $p = 0.033$).

dissociates from the spindle during anaphase. Our image analysis on cells expressing Alp7-neonGreen mCherry-Atb2 showed that the lack of Ppe1 results in significant reduction in the levels of Alp7 at the initial stages on spindle assembly (Fig 3E, 3F), consistent with the reduction of the overall levels of Alp7-3xGFP detected by western-blot on asynchronous cultures (Fig 3G, 3H). In the case of the $elp3\Delta$ mutant, although Alp7-3xGFP total levels were strongly reduced (Fig 3G, 3H), we could not detect a significant defect in Alp7-neonGreen accumulation at the mitotic spindle, compared to that of wild type cells (Fig 3E, 3H). Interestingly, our genetic analyses showed a very strong interaction of $ppe1\Delta$ with $alp7\Delta$, which impedes the suppression of the $cut7$–22 mutant by the double deletion of both $ppe1$ and $alp7$ (S3B Fig). This genetic interaction may be the consequence of reduced levels of specific proteins in the absence of Ppe1, which might be deleterious in the absence of Alp7. This result strongly suggests that the reduction in Alp7 levels is not the primary reason of the suppression of the $cut7$ phenotype by the absence of PP6 phosphatase.

Finally, we focused our attention to Ase1, the fission yeast PRC1 ortholog, involved in MT bundling [55], which was shown to be critical for the MT pushing-dependent spindle bipolarization in the absence of both Cut7 and Pkl1 [35]. The $ase1^+$ gene sequence shows no bias for the use of the AAA/ AAG codons for lysine (48% vs. 52%), although it has a strong bias for the usage of the GAA/ GAG codons for glutamate (76% vs. 24%. Paradoxically, although the inability to recruit Ase1 to the mitotic spindle was lethal in the $cut7\Delta$ $pkl1\Delta$ context [35], the absence of Ase1 completely suppressed $cut7$–22 growth defect at restrictive temperature (S3C Fig). Ase1 binds and stabilizes antiparallel MTs both in interphase and mitosis. Upon mitotic entry, cytoplasmic MTs catastrophe results in a drop of Ase1-GFP intensity adjacent to the SPBs. Nuclear Ase1 gradually associates to spindle MTs, rapidly accumulating in the mid-spindle during anaphase, to stabilize the MT overlap region. Indeed, the analysis of Ase1 levels in cells expressing Ase1-GFP showed a significant decrease of this protein at the spindle during mitotic progression in both $ppe1\Delta$ and $elp3\Delta$ cells compared to the wild type strain (Fig 3I, 3J). A strong reduction in the levels of Ase1–13Myc was detected by western-blot in asynchronous $ppe1\Delta$ and especially $elp3\Delta$ cells compared to wild type cells (Fig 3K, 3L).

Altogether, these results suggest that the reduction in the total levels of several MT regulators (at least Klp2 and Ase1) in *ppe1Δ* and *elp3Δ* cells, re-shape the force balance for mitotic spindle assembly, so that *cut7* mutants are competent for bipolar spindle assembly in such conditions.

**The absence of PP6 suppresses kinesin 5 mutant phenotypes independently of Elongator activity**

Ppe1 has been proposed as a positive regulator of the Elongator complex by dephosphorylating the serine S114 of the Elp4 subunit. Indeed, the expression of the hyperactive allele of *elp4*, Elp4-S114A, suppressed the sensitivity of *ppe1Δ* cells to rapamycin, derived from increased TORC1 activity [43]. Interestingly, from the genetic analysis performed throughout this study, we have noticed stronger implications derived from the lack of Ppe1 than those of the absence of Elongator activity complex (Figs 1A, 2A and 2D), suggesting that PP6 might play additional roles independently of Elongator. In fact, we found that *ppe1Δ* cells were more sensitive than *elp3Δ* cells to the presence of the aminoglycoside antibiotic geneticin G418, which affects protein translation by interacting with the 80S eukaryotic ribosome, increasing ribosome A-site tRNA miscoding [56] (S4A Fig). As expected, the *elp4-S114A* hyperactive allele of Elongator was resistant to the presence of this drug. Interestingly, the combination of *elp4-S114A* with *ppe1Δ* rescued *ppe1Δ* mutant sensitivity to G418, but did not behave as *elp4-S114A* cells (S4B Fig). This result confirms that PP6 has additional roles in protein translation to those of Elongator, yet Elongator may be a major target of this phosphatase and its hyperactivation may result in robust translation of many proteins affected by the lack of PP6 phosphatase.

Given that Elongator hyperactivation suppressed *ppe1Δ* sensitivity to G418 possibly by facilitating protein translation, we wondered if the expression of the *elp4-S114A* mutant would render *cut7–22 ppe1Δ* cells thermosensitive. Interestingly, the expression of *elp4-S114A* did not inhibit the suppression of the *cut7* mutant phenotype produced by the lack of Ppe1 (Fig 4A). This prompted us to examine if Klp2, Alp7 and Ase1 levels in *ppe1Δ* cells were restored to normal upon Elongator hyperactivation. Our analysis showed that Klp2, Alp7 or Ase1 levels in cells lacking Ppe1 were not significantly increased by the expression of the hyperactive *elp4-S114A* allele (Fig 4B-D). Altogether, these results suggest that PP6 phosphatase and the Elongator complex may share a common function in the translation of AAA and GAA-enriched transcripts, although this phosphatase might affect protein translation by other means. Moreover, the suppression of the *cut7* mutant phenotype might be due to the inability to produce specific proteins, since a global inhibition of protein translation does not support *cut7* mutant growth at restrictive temperature (S4C Fig).

Summing up, these results suggest that both PP6 phosphatase and Elongator complex ensure the proper translation of specific proteins involved in the force balance mechanism for spindle assembly. However, our data support the idea that PP6 phosphatase has additional roles to those of the Elongator complex in protein translation, which would explain the more robust suppression of the *cut7* phenotypes by the absence of Ppe1 than by the lack of a functional Elongator complex. It will be interesting to know which other proteins are specifically affected by PP6 and Elongator to untangle the roles of these complexes in different cellular processes.

## Discussion

Protein translation is a fundamental process for any living organism. This process is regulated at several levels to ensure cell homeostasis in a changing environment. An interesting control layer is exerted at the level of tRNA modifications, especially those performed to ensure optimal translation of mRNAs enriched in AAA codons (and GAA to a lesser extent). The Elongator complex, responsible for methyl-carbonylation of tRNAs at the uridine at the wobble position (U34), facilitates effective translation of such mRNAs. Previous work has revealed that the content of AAA codons in mRNAs is not random. Interestingly, genes involved in specific processes, such as oxidative stress response, cell differentiation, autophagy, cell cycle progression or subtelomeric gene silencing show a biased usage of AAA or GAA codons over their synonyms AAG or GAG, therefore making their translation partially dependent on Elongator complex function [43,54,57,58]. In this report, we show that Elongator also influences the expression of MT regulators that participate in spindle assembly.

PLOS Genetics

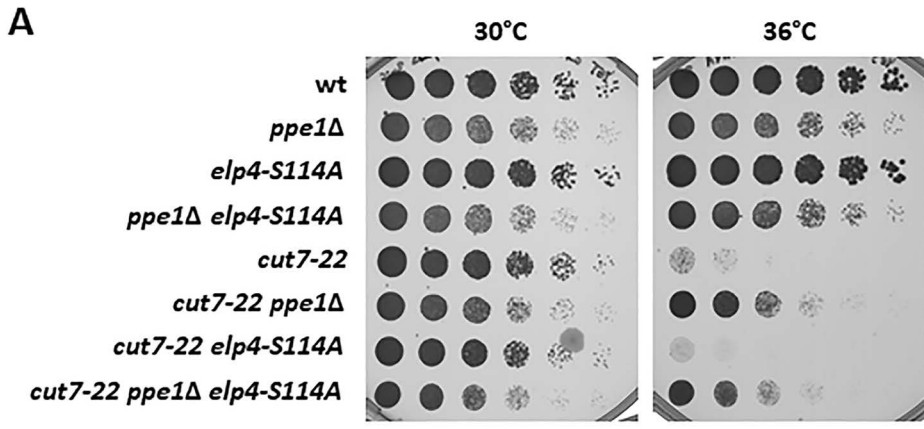

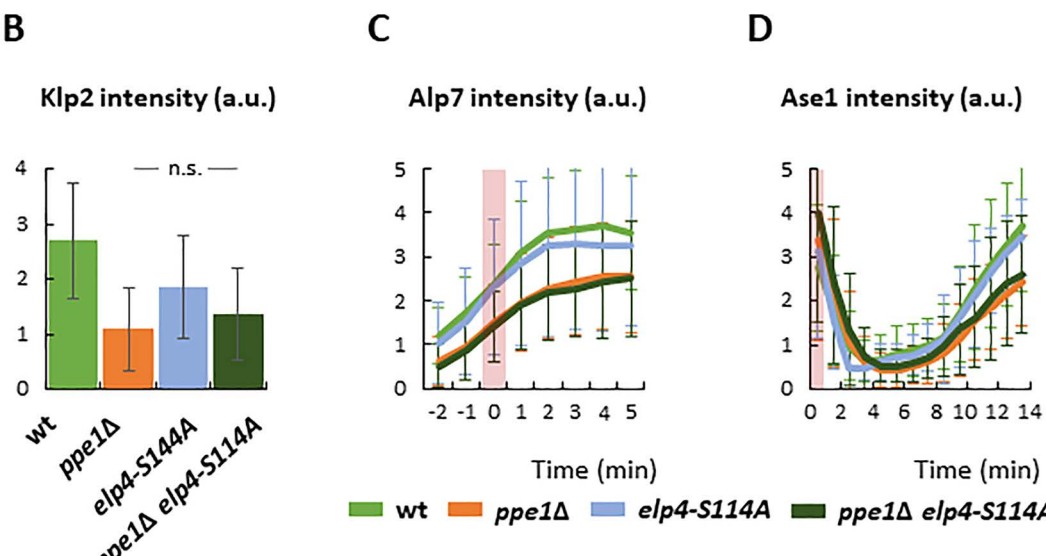

**Fig 4.  The absence of Ppe1 suppresses *cut7-22* thermo-sensitivity regardless of the hyperactivation of Elongator.**  A: Serial dilution assay of the wild type, *ppe1Δ*, *elp4-S114A*, *ppe1Δ elp4-S114A*, *cut7–22*, *cut7–22 ppe1Δ*, *cut7–22 elp4-S114A* and *cut7–22 ppe1Δ elp4-S114A* strains incubated for 3 days at the indicated temperatures. B: Plot showing the average of the Klp2-neonGreen signal intensity at the spindle region during the first three minutes upon mitotic entry. n for wild type: 27; n for *ppe1Δ*: 30; n for *elp4-S114A*: 29; n for *ppe1Δ elp4-S114A*: 30. a.u.: arbitrary units. Statistical significance was determined using a T-test (*ppe1Δ* vs. *ppe1Δ elp4-S114A*, p = 0.198). C: Plot showing the average accumulation of the Alp7-neonGreen signal intensity at the spindle region upon mitotic entry. The pink rectangle on the graph depicts the appearance of spindle MT. n for wild type: 51; n for *ppe1Δ*: 40; n for *elp4-S114A*: 30; n for *ppe1Δ elp4-S114A*: 30. a.u.: arbitrary units. Statistical significance was determined using a Repeated Measures ANOVA (*ppe1Δ* vs. *ppe1Δ elp4-S114A*, p = 0.664). D: Plot showing the average accumulation of the Ase1-GFP signal intensity at the spindle region upon mitotic entry. The pink rectangle on the graph depicts the appearance of spindle MT. n for wild type: 56; n for *ppe1Δ*: 59; n for *elp4-S114A*: 30; n for *ppe1Δ elp4-S114A*: 30. a.u.: arbitrary units. Statistical significance was determined using a Repeated Measures ANOVA (*ppe1Δ* vs. *ppe1Δ elp4-S114A*, p = 0.162).

Mitotic spindle assembly follows a dynamic force balance that, at initial stages permits spindle bipolarization (reviewed in [59]). Although several events contribute to the force balance (such as MT flux or chromosome capture), kinesins and dynein have a remarkably prominent role. Among these, kinesin 5, with its ability to crosslink and slide antiparallel MTs apart, is the major contributor to the production of outward pushing forces, required for separation of spindle poles at early

stages [3,4]. Indeed, the lack of the activity of this kinesin results in spindle collapse [15,21,23,60]. This phenotype can be suppressed to a certain extent by the inactivation of minus end-directed motors, dynein or kinesin 14 (depending on the organism) to counterbalance opposite forces [61,62]. These observations indicate that parallel pathways to kinesins 5, and 14 and dynein can compensate for their loss for spindle assembly, chromosome segregation and cell survival.

Over the last years, several proteins affecting MT dynamics have been involved in the regulation of the force balance for spindle assembly, highlighting the complexity of the process and the many layers of regulation that affect it. Among the hits of our suppressor screen on the fission yeast kinesin 5 *cut7–24* thermo-sensitive mutant, several clones contained mutations on regulatory proteins, such as PP6 phosphatase subunits and Polo kinase. We focused initially on PP6 mutants for two reasons: 1) the high frequency of appearance of mutated subunits of PP6 suggested an important role in spindle dynamics; 2) the relatively poor knowledge about the relationship between this phosphatase and mitosis.

Cut7, as other kinesin 5 members, may be regulated by different kinases, such as CDK, Aurora B or NIMA to regulate its targeting onto spindle MTs, its motor activity and possibly its directionality [15–17,20]. Given that it has been proposed that both Ppe1 and Ekc1 reside in the nucleus [63], we reasoned that Cut7 might be a target of PP6. Our data, however, suggests that this is not the case, since no change in Cut7 dynamic behavior during spindle assembly could be detected. Similar results were found for the opposing kinesin 14 protein, Pkl1. Altogether, these results suggested that PP6 suppresses *cut7* mutants defects independently of the regulation of the major players in the force balance mechanism, Cut7 and Pkl1.

In fact, previous work has suggested that Ppe1 and Ekc1 contribute to the activating dephosphorylation of the Elongator component Elp4 [43]. Our data indicates that defective Elongator activity, derived from either deleting any subunit of the complex, or of PP6 phosphatase subunits, may result in deregulation of the force balance for spindle assembly, which, in the context of kinesin 5 mutants, results in a partial bypass of efficient outward force production.

Indeed, protein levels of key MT regulators whose absence suppresses *cut7* mutants defects, such as Klp2, Alp7 [37], or Ase1, are decreased when Elongator is not active. The contribution of the reduction of each MT regulator in Elongator or Ppe1 mutants remains to be assigned. We reasoned that the suppression of the *cut7* mutant phenotype by the absence of Ppe1 should not be additive with the absence of the specific MT regulator, if they act on the same pathway. *ppe1* (and *elp3*) has a strong negative genetic interaction with *alp7*, suggesting that the reduction in Alp7 levels might not be the primary reason for Ppe1 suppression. Interestingly, the absence of Klp2 did not improve *ppe1* suppression of *cut7–22*, which places the reduction of Klp2 levels as a good explanation for this phenotype. The full deletion of *ase1* suppressed surprisingly well the *cut7* phenotype, even in the absence of Ppe1. This phenotype may be explained by the reduced MT crosslinking in such situation, which might favor MT outward pushing forces in the context of a partially inactive kinesin 5. It would be interesting to see if swapping the AAA and GAA codons by the synonym AAG and GAG in the genes coding for those MT regulators or, alternatively, overexpressing the corresponding tRNA would render *cut7* mutants thermo-sensitive independently of the presence of Elongator or PP6 phosphatase.

Recent work has revealed an unexpected role of Elongator in MT dynamics, specifically contributing to spindle asymmetry in Sensory Organ Precursor cells in *Drosophila*, essential for asymmetric cell fate determination [49]. In this system, Elongator localizes to spindle MTs, where it promotes MT growth and reduces MT catastrophe rate, increasing the overall MT lifetime [49]. On the light of this discovery, we investigated the hypothesis of a connection between *cut7* mutant suppression and a putative direct role of Elongator in controlling MTs in fission yeast. However, our results strongly suggest that Elongator is not a direct MT regulator in fission yeast. First, analysis of MT dynamics in *ppe1Δ*, *elp3Δ* or the catalytically inactive *elp3-YY531AA* mutant cells rules out this possibility. Secondly, Elp3 or Elp4 GFP fusions were not detected in cytoplasmic or spindle MTs. Altogether, we propose that defective protein production in Elongator and PP6 phosphatase mutants, but not the direct control of MT dynamics, is sufficient to suppress *cut7* mutants phenotypes. This also seems to be the cause of the synthetic phenotype shown with the SAC kinase mutant *mph1Δ*. Effective Elongator binding to MTs seems to require polyglutamylated tubulin dimers [64], a post-translational modification absent from fission yeast. This supports the idea that Elongator does not control MT dynamics directly in fission yeast.

Interestingly, our data suggest that Ppe1 not only affects Elongator for effective protein translation. In fact, the lack of Ppe1 repeatedly resulted in a milder reduction in MT regulator levels than that derived from the absence of the activity of Elongator. Moreover, the aminoglycoside antibiotic geneticin, which affects eukaryotic protein translation by increasing ribosome A-site tRNA miscoding [56], is more toxic to *ppe1Δ* than *elp3Δ* cells. Ppe1 and Ekc1 were recently detected as PP2A interacting partners, which, in turn, interacts with other tRNA-modifiers, as Trm112 [58]. It will be interesting to study additional roles of PP6 phosphatase in the control of protein production. However, although the expression of the *elp4* hyperactive mutant, *elp4-S114A*, did not restore Klp2, Alp7 or Ase1 levels in *ppe1Δ* cells, it did suppress *ppe1Δ* mutant sensitivity to geneticin. This result suggests that Elongator hyperactivation may result in a more robust translation of a subset of proteins, but PP6 phosphatase may regulate the production of others independently of Elongator. In line with this, *ppe1Δ* cells are much more sensitive to low temperature conditions, nutrient starvation or heat shock than Elongator mutants. The glycogen synthase kinase 3 Gsk3 ortholog has been shown to inhibit Elongator activity by phosphorylating Elp4-S114 [43]; it is therefore expected that the absence of this kinase would phenocopy the *elp4-S114A* mutant*,* unless Gsk3 had additional roles in the control of protein production. In the context of the force balance for spindle assembly, this may account for the more robust suppression of the *cut7* mutants phenotypes by *ppe1Δ* (which allows bipolar spindle assembly both in *cut7–22* and in the more stringent *cut7–24* mutant at restrictive temperature) as compared to *elp3Δ* (which can only suppress *cut7–22*).

Altogether, our data highlight that minor perturbations in MT dynamics, like the ones that result from a mild defective MT regulator protein translation, are sufficient to change the force balance for bipolar spindle assembly. In the context of transformed cells with deregulated cell cycle, where gene expression is highly altered, the unbalanced generation of force by altered protein levels may preclude the inability of spindle assembly upon kinesin 5 inactivation. Therefore, it becomes more important to unveil the contribution of different MT regulators to the force balance in order to design the proper strategy to impede uncontrolled cell division.

## Materials and methods

### Yeast genetics and culture

Standard *S. pombe* media and genetic manipulations were used [65]. All strains used in the study were isogenic to wild type 972 h- and are described in S2 Table. Strains from genetic crosses were selected by random spore analysis and replica in plates with the appropriate supplements or drugs.

Generally, cells were grown at 25 °C in YE5S before processing them either for microscopy or SDS PAGE analysis. For time-lapse experiments shown in Fig 1B and 2B, we noticed that *ppe1Δ* cells completely stopped their growth, not entering mitosis at all, during 8 hours after being transferred to 36 °C. Therefore, all strains except *cut7–22* cells were cultured overnight at 36 °C before imaging at this temperature. By contrast, *cut7–22* cells were cultured overnight at 25 °C and transferred to 36 °C for 3 hours before imaging.

For *nda3-KM311* synchronization experiments (S1C and S1F Fig), cells were cultivated overnight at 25°C in YE5S to an absorbance at 600 nm ($A_{600}$) of 0.6 and incubated for 8 hours at 18 °C before being processed.

DNA transformations were performed by using the lithium-DTT method. 20 ml of exponentially growing cells ($A_{600}$ 0.5-0.8) were harvested by centrifugation and washed with 10 mM Tris HCl pH 7.4. After a second centrifugation, they were re-suspended in 100 mM lithium acetate with 10 mM DTT and were incubated in an orbital wheel at room temperature for 40 minutes. 100 µl of these cells were mixed with 80µl of 100mM lithium acetate, 10 µl of single stranded DNA from salmon testes (D9156-5ML, Sigma) and 2 µg of the desired plasmid or the purified PCR product. After 10 minutes of incubation in an orbital wheel, 300 µl of PEG 4000, previously diluted 1:1 in 100 mM lithium acetate, were added. After a second round of 10 minutes on the wheel, 15 µl of DMSO were added and the cells were subjected to heat shock at 42 °C for 20 minutes in a water bath. Cells were then plated on the appropriate selection plates. Alternatively, an electroporation protocol was used. Briefly, 50 mL of exponentially growing cells ($A_{600}$ 0.5) were harvested by centrifugation and washed

first in cold sterile water and then in cold 1M sorbitol. Cells were resuspended in 500 μL 1M sorbitol and 40 μL of these cells were incubated in a pre-chilled electroporation cuvette with 500 μg of DNA resuspended in water. Cells received an electric pulse (1.5 kV for 6 ms) in a MicroPulser Electroporator (BioRad). After adding 900 μL of cold 1M sorbitol, cells were plated on the appropriate selection plates.

Drop assays were performed by serial dilutions of 1/4 from a starting sample of $A_{600}$ 1.0 of the indicated strains, which were plated on YE5S medium supplemented with the corresponding drug and incubated for 3 days at 30°C or 36°C, depending on the experiment.

## UV-radiation suppressor screen

1 x $10^7$ exponentially growing *cut7–24* cells were plated onto a total of ten YE5S plates, followed by exposure to sub-lethal amounts of UV-C (1 mJ/cm² of UV at a wavelength of 254 nm) using a Stratagene Stratalinker UV 1800 Crosslinker (Marshall Scientific). Plates were incubated for 5 days at 37 °C, conditions in which the untreated *cut7–24* strain produced no colony. 31 colonies grew in this condition. They were backcrossed to a wild type strain to discard possible revertants, rendering 28 *cut7–24* suppressors.

To identify suppressor mutations, genomic DNA of each suppressor was purified for NGS (Next Generation Sequencing). Briefly, exponentially growing cells were harvested by centrifugation, washed in 1 ml of sterile water and resuspended in 0.5 ml of cell wall digestion buffer (citrate/phosphate pH 5.2). Cell wall was digested by addition of 2 units of R-zymolyase 20T (Zymoresearch) and incubation at 37°C for 1 hour. Cells were lysed by addition of 0.3 ml of 20% SDS, followed by addition of 0.5 ml of chloroform and 1 min of vigorous mixing. Samples were centrifuged at 10.000 x g for 10 min, and the supernatants were transferred to Qiagen columns for DNA purification (QIAprep Spin Miniprep Kit). DNA was ethanol-washed and eluted in 10 mM Tris 1 mM EDTA. RNA was degraded by addition of 1 μl of 10 mg/ml RNAse A and incubation at 37 °C for 1 hour. 1.5 μg of DNA were submitted for NGS by Novogene Bioinformatics Technology (Hong Kong). The reads were mapped to the reference genome (Schizosaccharomyces_pombe.ASM294v2.genebank.gb). Unique indels and SNPs identified for each strain were reported. We manually mapped these mutations to the reference ORFs.

## Construction of mutant and tagged-protein strains

Deletion of *ppe1+*, *ekc1+*, *elp3+*, *elp4+*, *mph1+* and *klp2+* were created by PCR-amplification of the KanMX6 or NatMX6 antibiotic resistance cassettes from the respective pFA6a vector with the appropriate 100 nucleotides primers, as described in [66]. Wild type *cut7+*, *pkl1+*, *elp3+*, *elp4+*, *klp2+*, *alp7+* and *ase1+* were C-terminal fused to the green fluorescent proteins ENVY [39] or neonGreen [41], or to the 3xHA or 13xmyc tags at their C-terminus by PCR amplification of the tag and associated antibiotic resistance cassette from the respective pFA6a plasmids, and using the appropriate 100 nucleotides primers, as described in [66].

To create the *elp3-YY527AA* mutant, an integration module was created. Briefly, the integration plasmid for *elp3* constructs was produced by cloning of 500 pb of *elp3* promoter and 500 pb of *elp3* terminator into the pFA6a-GFP-kanMX6 [66], and using the restriction sites BamHI sites for the promoter sequence and SacI and SpeI for the terminator sequence. *elp3* ORF was cloned into the resulting plasmid by using the sites BamHI and AscI, with the reverse oligonucleotide containing the appropiate nucleotide sequence to create the amino acid substitutions Y527A and Y528A, replacing the *GFP* sequence from the original plasmid pFA6a-GFP-kanMX6, and generating the plasmid pSRP129. The resulting Elp3 mutant was termed Elp3-YY527AA, which includes the mutations on both tyrosine residues of the conserved tyrosine pair of *S. pombe* Elp3, known to be required for the acetyl and methyl transferase activity of Elongator complex [49,50].

To create the *elp4-S114A*, expressed from its endogenous *locus*, an integration cassette for *elp4* was built. For that purpose, 500 pb of *elp4* promoter and 500 pb of *elp4* terminator sequences were cloned into the pFA6a-GFP-kanMX6 plasmid [66], by using the sites SalI and BamHI restriction sites for the promoter and SacI and SacII for the terminator.

Then, *elp4-S114A* ORF was amplified from the genomic DNA of the strain yDH2005, which expresses Elp4-S114A-TAP, a kind gift from Dr. Damien Hermand [43], and cloned into the resulting plasmid by using the sites BamHI and AscI, replacing the *GFP* sequence from the original pFA6a-GFP-kanMX6 plasmid and generating the plasmid pSRP133.

All plasmids were checked by PCR and restriction enzyme digestion, and the DNA fragments amplified by PCR were sequenced.

## Microscopy techniques and image analysis

For time-lapse imaging, 300 µL of early log-phase cell cultures were placed in a well from a µ-Slide 8 well (Ibidi, Cat#80821) previously coated with 10 µL of 500 µg/mL soybean lectin (Sigma-Aldrich, Cat#L1395). Cells were kept for 1 min to attach to the bottom surface of the well and then, culture medium was removed carefully. Then, cells were washed three times with the same medium and, finally, 300 µL of fresh medium was added [67], before incubation in the microscope chamber at the same temperature as at which cells had been cultured.

Time-lapse images shown in Figs S1A, S1D, S2C, 3A, 3B and 3C are maximum intensity projections obtained from z-stacks of 7 slices at 1 µm intervals. Images were acquired every minute using an Olympus IX81 spinning disk confocal microscope with Roper technology controlled by Metamorph 7.7 software (Molecular Devices), equipped with a 100X/1.40 Plan Apo oil objective, a Yokogawa confocal unit, an EVOLVE CCD camera (Photometrics) and a laser bench with a 491–561 nm diode. Exposure time for green or red channels was 100 ms.

Intensity levels of the analyzed proteins shown in Figs S1A, S1D, S2C, 3A, 3B, 3C , 4B and 4C were performed by checking the intensity of the green channel in a circular region of 1 µm (for Klp2 and Ase1) or 2 µm (for Cut7, Pkl1 and Alp7) at the site of spindle MT nucleation at the stated times, and subtracting the intracellular intensity of a region of the same size placed at the cytoplasm of the analyzed cell.

Kymograph images shown in Figs 1B, 2B and S2B are maximum intensity projections, acquired by using a Dragonfly 200 Nikon Ti2-E spinning disk confocal microscope controlled by Fusion software (Andor), equipped with a 100X/1.45 Plan Apo oil objective, an Andor confocal unit, a sCMOS Sona 4.2B-11 camera (Andor) and a laser bench with a 405–561 nm diode (Andor). For Figs 1B and 2B, projections were obtained from z-stacks of 5 slices at 1.25 µm intervals. Images were captured every 2 minutes, with an exposure time of 0.3 s. For S2B Fig, cells were placed on a coverslip for image analysis; in this case, projections were obtained from z-stacks of 6 slices at 0.6 µm intervals, captured every 7 seconds and with an exposure time of 30 ms.

## SDS PAGE and Western-blot protein analysis

All protein extracts were obtained by TCA precipitation [68]. Briefly, 20 ml of cells cultured in YE5S to an OD of 0.6 were harvested by centrifugation. Cells were washed with 1 ml cold 20% TCA, resuspended in 100 µl cold 20% TCA, and lysed by mechanical beating with glass beads in a FastPrep FP120 (Savant Bio101). 400 uL cold 5% TCA were added and lysate was recovered by punching a hole on the bottom of the tube followed by centrifugation. After centrifuging for 5 minutes at 4000 x g, the supernatant was discarded, and the protein pellet was resuspended in 150 uL of resuspension buffer (2% SDS and 300 mM Tris Base). Samples were boiled for 5 minutes at 95°C, centrifuged at maximum speed and protein concentration of the clarified supernatant was measured by Bradford method.

5ug of total protein extract was separated by SDS-PAGE and transferred onto PVDF (polyvinylidene difluoride, Sigma-Aldrich Cat. No. 10600023). Membranes were incubated in blocking buffer (5% non-fat dried milk in TBST: 0.25% Tris, pH 7.6; 0.9% NaCl; and 0.25% Tween 20) for 1 h. Primary antibodies were monoclonal anti-HA (12CA5, Sigma-Aldrich; 1:10000), monoclonal anti-Myc antibody (9E10, Roche; 1:5000), monoclonal anti-GFP (7.1/ 13.1, Roche; 1:1000) and monoclonal anti-tubulin (B-5-1-2, Sigma-Aldrich; 1:10000). The secondary antibody was horseradish peroxidase-conjugated anti-mouse (Bio-Rad; 1:10000). The chemiluminescent signal was detected using the West-Pico Plus or

Femto ECL detection kit (Thermo-Fisher). The signal was captured with a Fusion FX6 system (Vilber) and quantified with ImageJ 1.54f software.

For PhosTag PAA gels, 40 μM PhosTag was added to 8.5% polyacrilamide. Gel electrophoresis and PVDF membrane transfer were performed as suggested by the supplier (Labmark PhosTag AAL-107).

### Statistical analysis

Sample size (n) is defined in each figure legend and derived from at least 3 independent experiments. The error bars correspond to standard deviation (SD) between experiments and are specifically indicated in each figure. P<0.05 were considered as statistically different results, analyzed by the Student's t test using Microsoft Excel and were denoted by an asterisk (*); non-statistically different results (P>0.05) were depicted by n.s. For protein accumulation profiles shown in Figs 1F, 1H, 3F, 3L, 4C and 4D, a Repeated Measures ANOVA was performed using R. P<0.05 were considered as statistically different results.

## Supporting information

**S1 Table. Table of strains used in this study.**
(DOCX)

**S2 Table. Supressor screen results.**
(XLSX)

**S1 Fig. Ppe1 does not control Cut7 or Pkl1 accumulation at the mitotic spindle. A:** Serial dilution assay of the wild type, *ekc1Δ*, *cut7–24* and *cut7–24 ekc1Δ* strains incubated for 3 days at the indicated temperatures. **B:** SDS PAGE analysis of Cut7-3xHA levels on wild type or *ppe1Δ* cells. The upper panel corresponds to the blot incubated with an anti-HA antibody, while the bottom panel corresponds to the blot incubated with an anti-tubulin antibody. **C:** Plot showing the total Cut7-3xHA levels relative to the corresponding tubulin levels in wild type and *ppe1Δ* cells. n for wild type: 3; n for *ppe1Δ*: 3. a.u.: arbitrary units. Statistical significance was determined using a T-test (p=0.086). **D:** Upper panel, PhosTag SDS PAGE analysis of Cut7-3xHA in asynchronous and mitotically blocked *nda3-KM311* cells in the presence or absence of Ppe1. Bottom panel, the same samples submitted to SDS PAGE analysis using an anti-tubulin antibody. **E:** SDS PAGE analysis of Pkl1-13xMyc levels on wild type or *ppe1Δ* cells. The upper panel corresponds to the blot incubated with an anti-Myc antibody, while the bottom panel corresponds to the blot incubated with an anti-tubulin antibody. **F:** Plot showing the total Pkl1-13xMyc levels relative to the corresponding tubulin levels in wild type and *ppe1Δ* cells. n for wild type: 3; n for *ppe1Δ*: 3. a.u.: arbitrary units. Statistical significance was determined using a T-test (p=0.121). **G:** Upper panel, PhosTag SDS PAGE analysis of Pkl1-13xMyc in asynchronous and mitotically blocked *nda3-KM311* cells in the presence or absence of Ppe1. Bottom panel, the same samples submitted to SDS PAGE analysis using an anti-tubulin antibody.
(TIF)

**S2 Fig. The absence of Elongator does not affect MT dynamics in fission yeast. A:** Serial dilution assay of the wild type, *elp3Δ*, *cut7–24*, *cut7–24 elp3Δ*, *cut7–22* and *cut7–22 elp3Δ* strains incubated for 3 days at the indicated temperatures. **B:** Selected kymographs showing interphasic MT dynamics in wild type, *ppe1Δ*, *elp3Δ* and *elp3-YY527AA* cells expressing GFP-Atb2, showing the inverted green channel in a grey scale. Scale bar: 5 μm. **C:** Plot showing MT growth rate of the indicated strains expressing GFP-Atb2 (n for wild type: 74; n for *ppe1Δ*: 87; n for *elp3Δ*: 79; n for *elp3-YY527AA*: 79). Statistical differences were determined using a T-test (wt vs. *ppe1Δ*, p=0.409; wt vs. *elp3Δ*, p=0.374; wt vs. *elp3-YY527AA*, p=0.001). **D:** Plot showing MT catastrophe rate of the indicated strains expressing GFP-Atb2 (n for wild type: 95; n for *ppe1Δ*: 73; n for *elp3Δ*: 82; n for *elp3-YY527AA*: 88). Statistical differences were determined using a T-test (wt vs. *ppe1Δ*, p=0.054; wt vs. *elp3Δ*, p=0.003; wt vs. *elp3-YY527AA*, p=0.294). **E:** Plot showing MT dwelling time

at the cell end of the indicated strains expressing GFP-Atb2 (n for wild type: 63; n for *ppe1Δ*: 67; n for *elp3Δ*: 67; n for *elp3-YY527AA*: 65). Statistical differences were determined using a T-test (wt vs. *ppe1Δ*, p = 0.931; wt vs. *elp3Δ*, p = 0.500; wt vs. *elp3-YY527AA*, p = 0.667). **F:** Maximum intensity projections from confocal time lapse images of wild type cells expressing mCherry-Atb2 and Elp3 or Elp4 fused to GFP. The upper series shows the magenta (mCherry-Atb2) and green (Elp3-GFP or Elp4-GFP) merged channels. The bottom series shows the inverted green channel in a grey scale. Time 0 corresponds to mitotic entry. The dashed line represents the cell contour. Scale bar: 2 µm. **G:** Western-blot analysis of SDS PAGE of Elp3-GFP and Elp4-GFP from the strains shown in C.
(XLSX)

**S3 Fig. Genetic interaction between *cut7–22, ppe1Δ* and the mutants of genes whose protein levels are affected in the absence of Ppe1 or Elongator activity. A:** Serial dilution assay of the wild type, *ppe1Δ*, *klp2Δ*, *ppe1Δ klp2Δ*, *cut7–22*, *cut7–22 ppe1Δ*, *cut7–22 klp2Δ* and *cut7–22 ppe1Δ klp2Δ* strains incubated for 3 days at the indicated temperatures. **B:** Serial dilution assay of the wild type, *ppe1Δ*, *alp7Δ*, *ppe1Δ alp7Δ*, *cut7–22*, *cut7–22 ppe1Δ*, *cut7–22 alp7Δ* and *cut7–22 ppe1Δ alp7Δ* strains incubated for 3 days at the indicated temperatures. **C:** Serial dilution assay of the wild type, *ppe1Δ*, *ase1Δ*, *ppe1Δ ase1Δ*, *cut7–22*, *cut7–22 ppe1Δ*, *cut7–22 ase1Δ* and *cut7–22 ppe1Δ ase1Δ* strains incubated for 3 days at the indicated temperatures.
(TIF)

**S4 Fig. The absence of Ppe1 suppresses *cut7–22* thermo-sensitivity regardless the hyperactivation of Elongator. A:** Serial dilution assay of the wild type, *ppe1Δ* and *elp3Δ* strains incubated for 3 days in YE5S of YE5S plates supplemented with 5 µg/ml of G418 at 30 °C. **B:** Serial dilution assay of the wild type, *ppe1Δ*, *elp4-S114A* and *ppe1Δ elp4-S114A* strains incubated for 3 days in YE5S of YE5S plates supplemented with 5 µg/ml of G418 at 30 °C. **C:** Serial dilution assay of the wild type, *cut7–22* and *cut7–24* strains incubated for 3 days at the indicated temperatures supplemented with 5 µg/ml of G418.
(TIF)

## Acknowledgments

We would like to acknowledge Damien Hermand, Ignacio Flor-Parra, Sergio Moreno, Rafael Daga, Pilar Pérez and Alicia García for kindly sharing strains and reagents. We thank the IBFG platforms that contributed to this work, especially Carmen Castro and Rebeca Martin (Microscopy Unit) for maintenance of the microscopes performed at the IBFG (Salamanca, Spain), and Jesús Pinto (Bioinformatics Unit) for his invaluable help assisting with the statistical analysis. We thank Deborah Bourc'his for the usage of the Stratalink UV crosslinker. We also thank David Martínez for his contribution in the initial moments of the project.

## Author contributions

**Conceptualization:** Javier Encinar del Dedo, Juan C. Ribas, Phong T. Tran, Sergio A. Rincon.

**Formal analysis:** Sergio A. Rincon.

**Funding acquisition:** Sergio A. Rincon, Juan C. Ribas.

**Investigation:** Laura Marín, Jorge Castro-Sangrador, Marta Hoya, Shara Tello, Pedro M. Coll, Alfonso Fernández-Álvarez, Sergio A. Rincon.

**Project administration:** Sergio A. Rincon.

**Supervision:** Sergio A. Rincon.

**Writing – original draft:** Sergio A. Rincon.

**Writing – review & editing:** Sergio A. Rincon.

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
