## [Decision Letter · Decision Letter 0]

16 Mar 2025

PGENETICS-D-25-00113

PP6 phosphatase and Elongator contribute to kinesin 5-dependent spindle assembly by controlling microtubule regulator levels

PLOS Genetics

Dear Dr. Rincon,

Thank you for submitting your manuscript to PLOS Genetics. After careful consideration, we feel that it has merit but does not fully meet PLOS Genetics's publication criteria as it currently stands. Therefore, we invite you to submit a revised version of the manuscript that addresses the points raised during the review process.

Please submit your revised manuscript within 60 days May 15 2025 11:59PM. If you will need more time than this to complete your revisions, please reply to this message or contact the journal office at plosgenetics@plos.org. Please include the following items when submitting your revised manuscript:

We look forward to receiving your revised manuscript.

Kind regards,

Alessia Buscaino

Academic Editor

PLOS Genetics

Pablo Wappner

Section Editor

PLOS Genetics

Aimée Dudley

Editor-in-Chief

PLOS Genetics

Anne Goriely

Editor-in-Chief

PLOS Genetics

**Journal Requirements:**

At this stage, the following Authors/Authors require contributions: Laura Marín. Please ensure that the full contributions of each author are acknowledged in the "Add/Edit/Remove Authors" section of our submission form.

The list of CRediT author contributions may be found here: https://journals.plos.org/plosgenetics/s/authorship#loc-author-contributions

3) We noticed that you used the phrase 'data not shown' in the manuscript. We do not allow these references, as the PLOS data access policy requires that all data be either published with the manuscript or made available in a publicly accessible database. Please amend the supplementary material to include the referenced data or remove the references.

5) We have noticed that you have uploaded Supporting Information files, but you have not included a complete list of legends. Please add a full list of legends for your Supporting Information files after the references list.

6) We note that your Data Availability Statement is currently as follows: "All relevant data are within the manuscript and its Supporting Information files.". Please confirm at this time whether or not your submission contains all raw data required to replicate the results of your study. Authors must share the “minimal data set” for their submission. PLOS defines the minimal data set to consist of the data required to replicate all study findings reported in the article, as well as related metadata and methods (https://journals.plos.org/plosone/s/data-availability#loc-minimal-data-set-definition).

7) Please amend your detailed Financial Disclosure statement. This is published with the article. It must therefore be completed in full sentences and contain the exact wording you wish to be published.

8) Please ensure that the funders and grant numbers match between the Financial Disclosure field and the Funding Information tab in your submission form. Note that the funders must be provided in the same order in both places as well. Currently, " J.C-S. and S.T. were supported by a predoctoral contract from the Junta de Castilla y León, funded by the European Social Fund Plus" is missing from the Funding Information tab.

**Reviewers' comments:**

Reviewer's Responses to Questions

Reviewer #1: This manuscript describes the results of a screen to suppress a mutation of a kinesin 5 in fission yeast (cut7-22). The screen identified the PP6 phosphatase and the authors explore the relationship between PP6 and kinesin 5. Their data point to an important role of PP6 and its target “elongator” – a six-protein complex involved in modifying tRNAs – with regulating spindle forces.

The experiments are clearly explained and the data most well presented. There is no direct mechanism uncovered (i.e. the specific PP6 targets), however, the ability of PP6 and elongator to suppress the kinesin mutant suggest that correct translational machinery are an important part of setting up spindle forced – at lease sufficiently so to cause lethality of the cut7-22 mutant.

I felt that the majority of the first three supplementary figures should be included in the main text. Significant sections of text refer to data in these figures, which is important for the progression of the experiments described. Since, there are currently only four main figures, I would recommend moving these supplementary figures into the main text. For example, the majority of the text from lines 321-334 is discussing data in Figure S3. I don’t think readers should have to delve into supplementary data, for figures that so strongly support the text. I am almost tempted to suggest inclusion of the screen results as a table in the main text – it is not huge and contains interesting other mutations e.g. Polo kinase (Plo1).

Figure 1C it should be possible to calculate a statistic (Fishers exact test?) to assure readers that the difference discussed in lines 176-177 is robust based on the numbers. It would be good to include the n numbers on the figure. This also applies to Figure 2C.

Figure 1D I assume that cu7-22 is not included in this figure, since mostly these cells do not form bipolar spindles (as per Figure 1C) – this should be mentioned.

The rescue to temperature tolerance of cut7-22 cells by elp4∆ is modest at best (but evident) – I would replace “partially” line 238, with “to a small extent”. The rescue for elp3∆ shown in the supplementary data is much clearer.

Figure 2D. I am struggling to see the synergy of the mutants. mph1∆ mutants are quite sensitive to MBC (especially at 3 µg/ml). When combined with elp3∆, elp3-YY527AA or ppe1∆, I am not convinced they are more sensitive than mph1∆ alone (note the double mutants grow poorly even on DMSO). Is there some way to quantify these more carefully?

I don’t see any statistical analysis to back up the conclusion that Alp7 intensity is reduced by 25% Figure 3B & line 310/311. I realise this reduction is not likely relevant for the cut7 mutat by PP6 mutations.

Figure 4B and 4C – do the statistical test back up that elp4-S114A “weakly increased” (line 365) the levels of Klp2 and Cls1 in a ppe1∆ mutant? I think it is “n.s.” for Klp2 and I don’t see a statistical test in Figure 4C.

Minor points

Line 39 “recently involved in” to “ recently been implicated in”

Line 153 fix “unable of assembling”

Line 227 “consequence” to “a consequence”

The strains list needs fixing; lots of highlighted text and a row labelled “supplementary figures”

Reviewer #2: kinesin-5 Cut7 is essential for mediating spindle bipolarization. By using a suppression screen, the authors found that the phosphatase PP6 (Ppe1 used in the study) and the elongator complex, which involved in the methyl-carbonylation of tRNAs during translation, exhibit epistasis interactions with cut7-22, a cut7 allele causing monopolar spindle at the restrictive temperature 37°C. Moreover, the authors demonstrate that PP6 and the elongator may be involved in regulating spindle bipolarization by modulating the translation of microtubule regulators, such as Klp2. Finally, PP6 appears to have additional roles other than the modulation of translation of microtubule regulators in regulating spindle polarization. The authors have a long-term track record in studying spindle bipolarization and discovered that spindle bipolarization is mediated by an antagonistic mechanism involved kinesin-5 and kinesin-12. Here, the authors advance the understanding of spindle bipolarization by showing that more types of proteins, i.e., the phosphatase PP6 and the elongator complex, are involved in the process. Therefore, the present work merits publication. Nonetheless, the following improvements may be considered during revision.

1) Too much background information was presented in the abstract. Perhaps, more results and conclusive statements, instead of background information, should be provided.

2) It is unclear how the absence of Ppe1 influences the protein level of Klp2.

3) It is unclear why the absence of either Alp7 or Ppe1 suppresses the effects of cut7-22 but double-deletion alp7 and ppe1 cannot.

4) Does the absence of Ppe1 influences the function of the elongator complex in fission yeast?

5) How do the authors explain the additive effects of the double deletion-mutants mph1-ppe1-deletion and mph1-elp3-deletion?

Reviewer #3: The paper by Marin et al. provides interesting new results showing

that PP6 phosphatase and the Elongator complex affect levels of

mitotic proteins that regulate microtubule dynamics and spindle force

in the fission yeast S. pombe. As a result, PP6 and Elongator protein

mutations are able to rescue defects in function of the kinesin-5

motor Cut7. Marin et al. therefore provide an important new

contribution to our understanding of the interplay of translation

regulation and mitotic function. The authors make a compelling case

that changes in translation of the MAPs/motors Klp2, Alp7, and Cls1

are the key mitotic consequence of perturbed Elongator function. This

paper is significant and I support its publication in PLoS Genetics. I

have some concerns about the interpretation of some results and the

rigor and reproducibility of the manuscript in its present form that I

recommend be addressed prior to publication.

Experimental results interpretation and rigor:

1. My major concern is the presentation and interpretation of gel and

blot results.

1a. The authors show subsets with only specific bands of the gels or

blots, and the molecular weight is indicated only by labels on the

bands with no markers shown. The authors should show the full gels and

blots for their results with the markers shown for a more rigorous

presentation. This will allow the reader to see, for example, whether

degradation products are present and how well their antibodies work.

1b. Further, the authors do not present total protein level

quantifications for their Western blots. These need quantification:

volume-to-volume comparisons are not sufficient because the yields may

vary during protein extraction. The authors should show

quantification, e.g. by BCA, to ensure that loading is equal. In the

manuscript the authors present tubulin as a loading control (for

example in figure S1B and other supplementary figures.) However, since

X-ray film was used, there is no way to know that they are in the

linear range of detection. Saturated exposure could mask loading

differences. Therefore in figures S1B and S1E for example, there could

be differences in Cut7 and Pkl1 expression that are not visible due to

saturated exposure. There are no data included to show that the X-ray

films that were scanned are in the linear range of detection. This is

a problem common to X-ray based quantification approaches. The authors

should address this.

1c. Related to the previous point, in figure S3 the tubulin bands

appear very different between A, B, and C. 2 bands are visible in A,

while only 1 is apparent in B and C. Can the authors explain the wildly

different appearance of their loading controls?

2. Phos-tag gel analysis.

2a. In lines 199-203 in the discussion of the Phos-tag gel analysis of

Cut7, the authors write "Our results showed an increased amount of

slow migrating Cut7 species in mitotic blocked cells compared to

asynchronous cells, suggesting that Cut7 may be phosphorylated during

mitosis. However, no differences in Cut7 migration were detected in

ppe1Δ cells, both during the mitotic block or asynchronous growth

(Figure S1C), reinforcing the idea that Cut7 may not be under direct

control of PP6 phosphatase." This interpretation of the results seems

over-simplified. A second band is visible in the low-temperature gel

that appears to be absent in ppe1Δ. There also is a shift in of the

18 degree band relative to the 28 degree band in the phosphatase

deletion. It's perplexing result that with removal of a phosphatase

you lose bands. How do the authors explain this? The gel as

presented also suggests there is a difference between the running size

in the 28 degree condition because in the ppe1Δ is running lower. I

suggest that this be discussed more extensively. For real comparison

between the 28 deg bands of both strains should be run next to each

other.

Further, the percentage of the Phos-tag gels is not given. I suggest

that the authors include this information. I also recommend that they

run higher percentage gels to resolve these apparent differences.

2b. In figure S1F why does Pkl1 look so different in the Phos-tag gel

from Cut7? It appears that the dumbbelling observed in the lanes would

indicate that the lanes are over-loaded. And for the reasons described

above, it's impossible to know if there is the same amount of protein

in each lane because (1) there is no loading control shown and (2)

apparent size differences could simply be caused or masked by loading

differences.

3. CLASP results. In lines 325-327, the authors state "Cls1 is an

essential protein for spindle assembly ((56)), therefore we used a

truncation of the PRC1 ortholog, ase1Δ(665-731), which lacks the

interacting region with Cls1 and fails to recruit it properly to the

mitotic spindle ((55))." The dependence of CLASP recruitment on Ase1

binding is controversial; contradictory results have shown that

interaction with Ase1 is not required for Cls1 recruitment (Ebina et

al Biology Open 2019). Other labs have also observed that Ase1 is not

required for Cls1 recruitment in unpublished results. Therefore, the

ase1Δ(665-731) strain not the best choice to perturb CLASP binding to

spindle microtubules. The authors could consider a different strategy

to perturb CLASP, but at least should discuss the interpretation of

this result.

4. In lines 435-439 the authors refer to global phosphoproteomics

analysis in the PP6 deletion. I did not find any data on these

experiments in the paper. They should be presented.

Writing and minor suggestions. I suggest that the authors consider

these minor points to improve the presentation of their results.

5. The introduction presents a thorough and clear description of

fission-yeast mitosis and the roles of Cut7, Pkl1, and other spindle

proteins. However, there is only one paragraph that briefly discusses

PP6 and Elongator, with the fission-yeast gene names for these

proteins not mentioned. I believe that most readers of the paper will

likely be more familiar with pombe mitotic proteins than with the

roles of PP6 and Elongator in translation regulation, so more

introduction to this process and proteins would be very helpful to the

reader. I did not immediately recall, for example, the Elongator

background presented in lines 222-233 and this made the paper more

difficult to follow. Information on the importance of codon bias of

AAA versus AAG for lysine as a potential contributor to perturbation

of Elongator function is not presented until lines 299-300. I

recommend that the authors discuss this more thoroughly in the

introduction. A schematic of what's known about PP6 and Elongator

early in the paper would also be helpful.

6. In the discussion in lines 424-434 the authors claim that the

Phos-tag gels rule out a direct role of PP6 in Cut7 dephosphorylation

seems to be to be an over-interpretation.

7. The cut7-24 mutation site discussed in the intro but the same is

not done for cut7-22.

8. I find it interesting that cut7-22 has more robust suppression of

the ppe1 deletion compared to cut7-24; this result is even more

dramatic in the elp3 deletion. What do the authors make of this? It

would be interesting to discuss.

9. The order of presentation is confusing in several places. Figure 1E

is discussed before B-D; why not move this panel earlier in the

figure? In several places supplementary figures are discussed before

main figures, for example S4 discussed before 4. In addition figure

S3F discussed before S3C and S3E. I suggest changing either the figure panel

or discussion order.

10. The authors should check their use of the word astringent on line

501; perhaps they mean stringent?

11. In the supplement that caption of figure S2 is given after the

caption of figure S3.

**Have all data underlying the figures and results presented in the manuscript been provided?**

Reviewer #1: Yes

Reviewer #2: Yes

Reviewer #3: Yes

PLOS authors have the option to publish the peer review history of their article (what does this mean? ). If published, this will include your full peer review and any attached files.

**Do you want your identity to be public for this peer review?** For information about this choice, including consent withdrawal, please see our Privacy Policy .

Reviewer #1: No

Reviewer #2: No

Reviewer #3: No

**Figure resubmission:**
---

## [Decision Letter · Decision Letter 1]

24 Sep 2025

Dear Dr Rincon,

We are pleased to inform you that your manuscript entitled "PP6 phosphatase and Elongator contribute to kinesin 5-dependent spindle assembly by controlling microtubule regulators levels" has been editorially accepted for publication in PLOS Genetics. Congratulations!

Yours sincerely,

Alessia Buscaino

Academic Editor

PLOS Genetics

Pablo Wappner

Section Editor

PLOS Genetics

Aimée Dudley

Editor-in-Chief

PLOS Genetics

Anne Goriely

Editor-in-Chief

PLOS Genetics

BlueSky: @plos.bsky.social

Comments from the reviewers (if applicable):

Reviewer's Responses to Questions

**Comments to the Authors:**

Reviewer #1: The authors have addressed each of my concerns, raised in the original review. I am happy that the changes made improve the manuscript and support the conclusions made.

**Have all data underlying the figures and results presented in the manuscript been provided?**

Reviewer #1: Yes

PLOS authors have the option to publish the peer review history of their article (what does this mean? ). If published, this will include your full peer review and any attached files.

**Do you want your identity to be public for this peer review?** For information about this choice, including consent withdrawal, please see our Privacy Policy .

Reviewer #1: No

**Data Deposition**

http://datadryad.org/submit?journalID=pgenetics&manu=PGENETICS-D-25-00113R1

**Press Queries**

---

## [Editor Report · Acceptance letter]

PGENETICS-D-25-00113R1

PP6 phosphatase and Elongator contribute to kinesin 5-dependent spindle assembly by controlling microtubule regulators levels

Dear Dr Rincon,

We are pleased to inform you that your manuscript entitled "PP6 phosphatase and Elongator contribute to kinesin 5-dependent spindle assembly by controlling microtubule regulators levels" has been formally accepted for publication in PLOS Genetics! Your manuscript is now with our production department and you will be notified of the publication date in due course.

With kind regards,

Lilla Horvath

PLOS Genetics

On behalf of:
